# An Ecological Study Relating the SARS-CoV-2 Epidemiology with Health-Related, Socio-Demographic, and Geographical Characteristics in South Tyrol (Italy)

**DOI:** 10.3390/ijerph21121604

**Published:** 2024-11-30

**Authors:** Antonio Lorenzon, Lucia Palandri, Francesco Uguzzoni, Catalina Doina Cristofor, Filippo Lozza, Cristiana Rizzi, Riccardo Poluzzi, Pierpaolo Bertoli, Florian Zerzer, Elena Righi

**Affiliations:** 1Epidemiological Surveillance Unit, South Tyrolean Healthcare Agency, 39100 Bolzano, Italy; catalinadoina.cristofor@sabes.it (C.D.C.); f.lozza001@unibs.it (F.L.); riccardo.poluzzi@ior.it (R.P.); pierpaolo.bertoli@sabes.it (P.B.);; 2Department of Biomedical, Metabolic and Neural Sciences, University of Modena and Reggio Emilia, 41125 Modena, Italyelena.righi@unimore.it (E.R.); 3Clinical and Experimental Medicine PhD Program, University of Modena and Reggio Emilia, 41125 Modena, Italy

**Keywords:** SARS-CoV-2, herd immunity, health inequities

## Abstract

The literature associating the spread of SARS-CoV-2 with the healthcare-related, geographical, and demographic characteristics of the territory is inconclusive and contrasting. We studied these relationships during winter 2021/2022 in South Tyrol, a multicultural Italian alpine province, performing an ecological study based on the 20 districts of the area. Data about incidence, hospitalization, and death between November 2021 and February 2022 were collected and associated to territorial variables via bivariate analyses and multivariate regressions. Both exposure variables and outcomes varied widely among districts. Incidence was found to be mainly predicted by vaccination coverage (negative correlation). Mortality and ICU admission rates partially followed this distribution, while the case fatality rate was inversely correlated to average salary, and hospital admission rates increased where hospitals capacity was higher, and from the southern to the northern border of the province. These findings, besides confirming the efficacy of vaccination in preventing both new and severe SARS-CoV-2 cases, highlight that several geographical and socio-demographic variables can be related to disease epidemiology. Remote areas with wage gaps and lower access to care suffered most from the pandemic. Our findings, therefore, underly the existence of health inequity issues that need to be targeted by implementing specifically tailored public health interventions.

## 1. Introduction

The SARS-CoV-2 pandemic has dramatically affected our planet in its entirety, with the total amount of registered cases reaching about 250 million by November 2021, just a year and a half after the pandemic began [1]. Due to the arrival of new variants, this figure even tripled by the end of the same winter [2]. The impact on people’s health was immense, with 5 million detected deaths rising to almost 7 during the same period [1]. Although the virus reached every continent, its spreading varied significantly in different geographical habitats, seasons, and communities [3,4,5], and factors related to this phenomenon are still under study worldwide [6,7].

In Italy, for example, in the northernmost province of Bolzano, also called South Tyrol, the SARS-CoV-2 pandemic showed epidemiological characteristics differing greatly both from the neighboring territories and across its territory, notably during winter 2021/2022, such as a higher incidence and lower vaccination adherence [8,9]. This area has also peculiar and heterogeneous geo-morphological and socio-cultural characteristics. Its natural environment resembles that of further northern countries, although the peaks of the southern Alpine slope protect it from the cold polar currents, giving it a sunny Mediterranean climate, especially in its southern part bordering other Italian regions [10]. Further, its multifaceted historic background is reflected in the great socio-cultural heterogenicity of its 534,912 inhabitants, with 65% of the population being German-speaking and evenly distributed across the territory; 30% Italian-speaking and concentrated in the low-altitude cities; and 5% Ladin, a local ancient population of the higher mountains, although there is considerable mixing between groups [11].

South Tyrol is administered as an ‘autonomous statute region’ and divided into 20 health districts grouped into four sections built up around the four largest municipalities in the area (Bolzano, Merano, Bressanone, and Brunico), where the main health services are located [11,12], as represented in Figure 1. The section of Bolzano, the densely populated capital city with 107,467 inhabitants, includes both the southern, low-lying districts, with a strong Italian cultural influence, and those higher up, more anchored in the German or Ladin tradition. Around Merano, the districts are mostly composed of isolated rural sunny valleys devoted to agriculture and farms. The Bressanone section is a predominantly mountain area, where low tourism is balanced by high trade, thanks to the presence of the main link between Italy and Austria. Finally, the Brunico section is characterized by isolated and highly touristy valleys, which are important links with Austria and the other Italian Dolomite regions [12].

The literature investigating the epidemiological variability of the disease related to health services provision and vaccination coverage (herd immunity) and to the geographical or cultural characteristics that mark this territory, such as altitude, tourism, and population dynamics, is scarce, and evidence is conflicting [9,14,15,16,17,18]. Therefore, the aim of this study is to investigate whether the deep geographical heterogeneity of socio-cultural, morphologic, and health-related features characterizing the South Tyrol territory (northern Italy) can be related to the differing spread of SARS-CoV-2 observed across the region in the first winter of the pandemic characterized by the absence of social restrictions and the availability of effective preventive measures, including vaccination.

## 2. Materials and Methods

We carried out an ecological study based on the 20 districts composing the Province of Bolzano (South Tyrol), Italy. Aggregated data on geo-morphological, socio-cultural, and health service-related features and SARS-CoV-2 cases and deaths were collected from the Local Health Authority (SABES) [19] or the Provincial Institute of Statistics (ASTAT) [20].

Geographical, socio-demographic, and health service-related variables of the area were the exposure variables of interest, while SARS-CoV-2 epidemiological measures of occurrence (incidence, hospitalization, or death) were the outcomes evaluated.

The study protocol was approved by the Bolzano Hospital Ethics Committee on 16 March 2022 (Prot. 0259655-BZ).

### 2.1. Exposure Variables

Geo-morphological, socio-cultural, and health service-related data were taken from the public ASTAT database (accessed in April 2024), referring to the most recent period available for each variable [20]. Information on vaccination coverage was provided by the Local Health Authority [19].

The following data were available.

Demographics:Inhabitants, represented by the resident population amount (2021);Population density (people living in every square km, 2021).

Social dynamics variables:Average yearly salary, for employees in the private sector (2019, before the pandemic);Winter tourism, expressed as the number of tourists from outside the province who overnighted at least once in any type of local accommodation (November–April 2019).

Geographic variables:Presence of main cities;Average altitude, or vertical distance above the reference sea level (meters, current);Bordering other Italian regions through winter-opened mountain passes or ordinary roads (current);Bordering other countries (Austria and Switzerland) through the same roads (current).

Health Services-related variables:Number of pharmacies every square km (2010);Presence of small (without an intensive care unit—ICU) or big-sized (with ICU) hospitals (current) [19];Primary series vaccination coverage, updated as of 12 February 2022 and expressed as the number of subjects who received primary vaccination reported for the total resident population (%). In this variable, we included subjects with one dose of viral vector vaccine, two of the mRNA vaccine, or one of any with a previous infection, in the timeframe and with the vaccines authorized by the Italian Health Ministry [21,22];Booster/additional dose coverage, a variable including the number of subjects who received a booster (or additional) dose of vaccine with the serums defined by the Italian Health Ministry, reported for the total resident population, updated as of 12 February 2022, and expressed as a percentage [23]. The booster vaccination campaign in South Tyrol began in November 2021 and was initially reserved for the most vulnerable. The vaccination was available at general practitioners and major public health facilities of the province.

### 2.2. Outcome Variables

Health data on SARS-CoV-2 cases referring to winter season 2021/2022 were provided by the Epidemiological Surveillance Unit of Bolzano that oversaw tracking SARS-CoV-2 cases in the entire territory of South Tyrol [11].

The following information was available.
SARS-CoV-2 cases: the total number of all new cases occurring in the resident population between 1 November 2021 and 12 February 2022. To this category belong all patients who had taken a diagnostic swab with a positive result in any authorized center of the province without a recorded positive result in the previous 90 days [24].Hospitalizations: total number of hospitalizations related to the SARS-CoV-2 cases occurring up to 21 days (lastly up to 4 March 2022) after diagnosis [25]. Total hospitalizations for SARS-CoV-2 and ICU admissions were considered. Admissions were excluded if referring to patients who were positive but asymptomatic for COVID and suffered from diseases unrelated to the infection.Deaths: total number of deaths caused or contributed to by SARS-CoV-2, linked to the cases included in the study (lastly up to 4 March 2022) after diagnosis.

### 2.3. Data Processing

ASTAT data were displayed by municipality, so an aggregation of data at the district level was performed to make them comparable with district-related outcome data, following the belongings of each municipality to the 20 districts, as depicted in Appendix A.

A sum was made for the numeric variables, such as winter tourists, inhabitants, and surface. A weighted average for the number of inhabitants was performed to assess districts’ average salary. Average altitude, pharmacies density, and population density were weighted for the surface size, while the highest municipality value for hospital presence and size, borders information, and main cities was used for the district.

Outcome amounts were clustered for each district and divided into epidemiological measures.
Incidence of SARS-CoV-2 during the whole winter 2021/2022 was calculated for each district, relating the cases amount to the number of inhabitants and expressed per 100,000 inhabitants.Hospitalization Rates were assessed in the same way, relating both ordinary and ICU admissions to the inhabitant amount of each district and reported for every 100,000 inhabitants.Hospitalization proportion corresponded with the number of both ordinary and ICU admissions related to the number of cases in the same district and expressed per 10,000 cases.Mortality and case fatality rate (CFR) were assessed relating the district’s death amount per 100,000 inhabitants and per 10,000 cases, respectively.

### 2.4. Statistical Analyses

The variables were first summarized by descriptive analyses, and then correlations and bivariate non-parametric statistical analyses were performed (Spearman’s correlation, Kruskal–Wallis, and Mann–Whitney U tests) to assess the relationships between the variables.

The variables associated with each outcome (*p*-value < 0.1) were then analyzed together by multiple linear regression, performing the model for the outcomes with at least three non-collinear variables associated with it. An easier interpretable scale for the exposure variable was used in the model: population density was considered for inhabitants per hectare (10,000 m^2^), winter tourist amount expressed by 100,000 units, and altitude expressed in kilometers. Variables with VIF (Variance Inflation Factor) > 5 were considered collinear and not included in the linear regression. The distribution of residuals and Nagelkerke’s adjusted R square were evaluated to ensure a good fit of the model to the data.

Statistical analyses were performed using Jamovi software (version 2.5.5) [26].

## 3. Results

### 3.1. Main Features of the Area

The 20 districts in the province of Bolzano differ greatly in geo-morphological and socio-cultural characteristics and health-care facilities distributions, as shown in Figure 2 and Appendix A.

Four districts are directly connected with Austria, six with other Italian regions. Laives-Bronzolo-Vadena (LBV) has the lowest altitude (249 m to the sea level), Val Gardena the highest (1454 m), and the median of the province is 961 m (IQR of 338).

The number of inhabitants (district median: 20,259; IQR: 11,536) ranges from less than 10,000 in Val Passiria to over 100,000 in Bolzano, that of winter tourists (median: 491,182; IQR: 701,423) from 78,618 in LBV to 1.7 million in Val Badia. The lowest wage in the private sector is in Val Passiria (25,591 euros/year), the highest in Brunico (29,132 euros/year).

Big hospitals are located in the four main cities (Bolzano, Merano, Bressanone, Brunico) and small ones in Media Val Venosta, Alta Val Pusteria, and Alta Valle Isarco. Pharmacies for each district are a median of 1 per every 100 square km. At winter 2021/2022 end, primary series vaccination coverage varied extremely, with an IQR between 68% and 72% (median: 70%); booster dose coverage was 43–46% (median 45%). The lowest values are in Val Passiria (57% primary series, 31% booster), the highest in Bolzano (75% and 51%).

The geographical distribution of the SARS-CoV-2 pandemic is shown in Figure 2 and Appendix A. Outcomes varied greatly across districts. The SARS-CoV-2 incidence was lowest in Bolzano and highest in Val Gardena, with an overall district median of 18,090 cases per 100,000 inhabitants (IQR: 2356). The hospitalization rate (per 100,000 inhabitants) ranged from 90 in Bassa Atesina to 256 in Brunico (district median: 162; IQR: 48). The ICU admissions rate median was 15 per 100,000 inhabitants (IQR: 4–23). The district median of CFR was 19 per 10,000 cases (IQR: 4), ranging between 5 in Val Gardena and 31 in Media Venosta.

### 3.2. Territorial Characteristics and SARS-CoV-2 Epidemiology

According to bivariate analyses (Table 1 and Table 2), the SARS-CoV-2 incidence at the district level appeared significantly correlated to several district factors and features, including the total number of inhabitants and of winter tourists, the average altitude, and both primary and booster dose coverage. On the contrary, the distribution of healthcare facilities such as pharmacies or hospitals was not associated with any significant variation in this parameter.

Both the hospitalization rate and the proportion of hospitalization cases resulted in being significantly higher in districts with main hospitals and cities and in districts bordering other countries or not bordering other Italian regions. District ICU hospitalization, the proportion of ICU cases, mortality, and CFR resulted in being less related to the ecological variables analyses in this study, excluding population density for the ICU admission rate or proportion of cases or average salary for CFR.

The results of the multiple linear regressions, carried out with the incidence or hospitalizations proportion as the outcome, are shown in Table 3. Due to collinearity issues, a limited number of variables could be included in the models (Appendix A).

SARS-CoV-2 incidence was mainly related to vaccination coverage (*p*-value = 0.033), while the linear regression for hospitalizations identified geographical location and the presence and size of hospitals as the most statistically reliable predictors. The distribution of the variables included in the second model and of the outcome is represented graphically in Figure 3.

## 4. Discussion

During winter 2021/2022, SARS-CoV-2 was diagnosed in almost one out of every six people in Bolzano province (South Tyrol). However, the incidence was unevenly distributed across the territory, reflecting its complexity. According to our ecological study, the variable most related with its distribution across districts was the vaccination coverage [9]. In addition, different geographical and socio-demographic features of the territory appeared to play a role in the spread of the disease. Namely, the incidence was higher in districts where population density was greater, in traditional mountain areas, and in those with higher tourist flows.

ICU admissions and mortality due to SARS-CoV-2 partially tracked the incidence, being higher at higher altitudes and in districts with less vaccination coverage, while higher rates of overall hospitalization were observed in the districts bordering other countries in the north, and lower rates in those bordering the Italian regions to the south. Moreover, hospitalization was highest where the capacity of healthcare facilities was the greatest, i.e., in major cities, where major hospitals are located, and the lowest in districts where hospitals are absent. Case fatality rate (CFR), on the other hand, was found to be negatively correlated with the average district salary.

The results of this study can be evaluated in the light of the existing literature, even though comparisons can be difficult due to the scarce studies analyzing similar variables often reporting contrasting results [7,27]. The role played by altitude, for example, is still unclear, as it has been found in previous studies to be both negatively and positively associated with the spread of the disease [14], and this relationship is only partially supported by pathophysiological hypotheses [28]. Population density and number of inhabitants, on the other hand, resulted in being almost always positively associated with incidence, since a larger and more concentrated population increases human contact and consequently also the likelihood of contagion. In our study, we found similar correlations; nevertheless, the strength of these relationships fades when performing multivariate analyses.

Indeed, vaccination coverage was confirmed to be the most relevant predictor of the geographical distribution of SARS-CoV-2 incidence, being also associated with intensive care admissions and deaths. Although this is one of the first studies suggesting the presence of a herd immunity effect at the ecological level, a lower prevalence of the disease had already been predicted at a higher vaccination coverage [18], and the effectiveness of vaccination at the individual level has been well established in preventing severe SARS-CoV-2 disease [29,30], but also the contagion, both through increased protection against infection [31] and less contagiousness [32]. The lower vaccination coverage could be therefore the underlying reason of the increased incidence of the disease in the less inhabited and higher districts, geographically and culturally distant from the health governance authorities, which have very little trust placed both in them and their promoted public health interventions, such as vaccines.

Finally, we analyzed the role played by winter tourism, which is an important part of the local economy and was considered responsible for a significant spread of the virus in the first pandemic wave [17]. However, our study highlights how during the first winter without lockdowns, tourism contributed to a very limited extent to the spread of the disease in the resident population, even not followed by an increase in the severe form. This is probably due to an effective implementation of the social and community preventive measures, such as isolation of positive people, social distancing, personal protection equipment (PPE), made mandatory during that winter. Effective public health training and communication might prove to be crucial for managing outbreaks [33,34]. Clear communication on prevention and vaccination might help build trust and compliance with public health measures.

Geographical and socio-demographic features of this territory seem to have a greater influence on the hospital admissions, which did not follow exactly the distribution of incidence, ICU admissions, and mortality, and which resulted in being linked also to other variables, such as geographical location (higher rates were observed on the northern border with other countries, lower in the south close to the other Italian regions) and the accommodation capacity of health facilities. If, for the former association, we can assume a protective role of the milder and sunnier climate in the southern districts of the province [35], as already shown by other reliable studies [4,5], the second association strongly suggests that living in a district with highly responsive healthcare structures facilitates access to it, underlying the potential of health inequalities. The possible presence of health disparities is further highlighted by the negative correlation between the CFR of infection and the average income of the district.

Finally, a regional infectious disease surveillance system is vital for monitoring and controlling disease spread. It allows for the timely collection and analysis of data, helping to identify outbreaks early and implement targeted interventions [36,37]. In Bolzano province, localized surveillance revealed how factors like vaccination coverage and population density influenced SARS-CoV-2 incidence. This enables efficient resource allocation and tailored public health measures, reducing disease burden.

This article has considerable strengths, such as the possibility of including all SARS-CoV-2 cases detected during the period of interest by the surveillance system implemented by the Local Health Authority and of analyzing a very wide and varied range of geographical, socio-cultural, and health-related variables in the area studied. Nevertheless, there are also some important limitations, first of all that of analyzing only a specific period of the pandemic (winter 2021/2022), without therefore considering the previous waves, the deaths that had already occurred, and their distribution, although the total number of cases recorded in the province up to that time we considered was less than a third of that at the end of the period studied [38].

Furthermore, a possible under-diagnosis could have occurred, especially in the more vaccinated district, since during the period under study, only non-vaccinated subjects had to routinely test negative to obtain a green pass to work or participate in most social activities [39]. Consequently, this could have led to the detection of a higher number of asymptomatic cases in districts with less vaccination coverage, explaining the higher incidence. However, the ICU admissions rate and mortality distribution showed similar correlations patterns of incidence rate, and similar associations patterns were observed when considering cases or inhabitants as rate denominators; all these findings suggest a uniform detection of cases across the territory. Furthermore, the number of SARS-CoV-2-positive hospitalizations and deaths data should be really accurate, since all hospitalized and deceased subjects were tested, and no private hospitals for acute diseases are found in South Tyrol.

Finally, ecological bias cannot be ruled out, as the variability of SARS-CoV-2 epidemics in South Tyrol is driven also by several underlying individual-level conditions. Further studies based on individuals are therefore needed to investigate deeper the main findings emerging from this study.

## 5. Conclusions

This study, by analyzing the distribution of SARS-CoV-2 and its outcomes during a high-incidence period in the multifaceted territory of South Tyrol, shows that, besides the effective protective effect played by vaccination coverage, several geographical and socio-cultural variables can be associated with the disease incidence, hospitalization, and mortality. Overall, more remote and higher-altitude areas with less availability of health facilities suffered the most, confirming the need of targeting specific subgroups and geographical areas by implementing specific territorial-tailored public health interventions to address more efficiently potential heath inequities and reduce social and economic health-related costs both to individuals and society.

## Figures and Tables

**Figure 1 ijerph-21-01604-f001:**
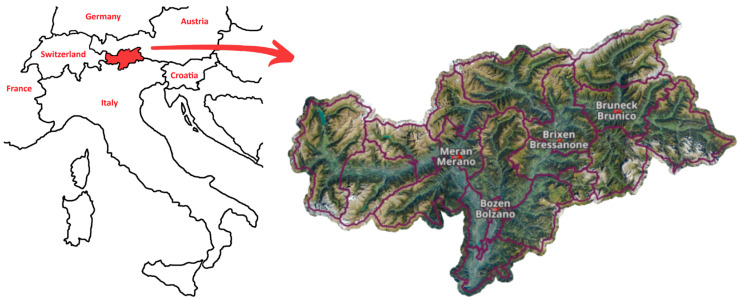
Geographical representation of South Tyrol’s position in relation to Italy and neighboring countries (**left**) and physical map of the province divided into the 20 health districts with the four main section centers (**right**). Image adapted from the online public geographic data tool of the Autonomous Province of Bolzano [13].

**Figure 2 ijerph-21-01604-f002:**
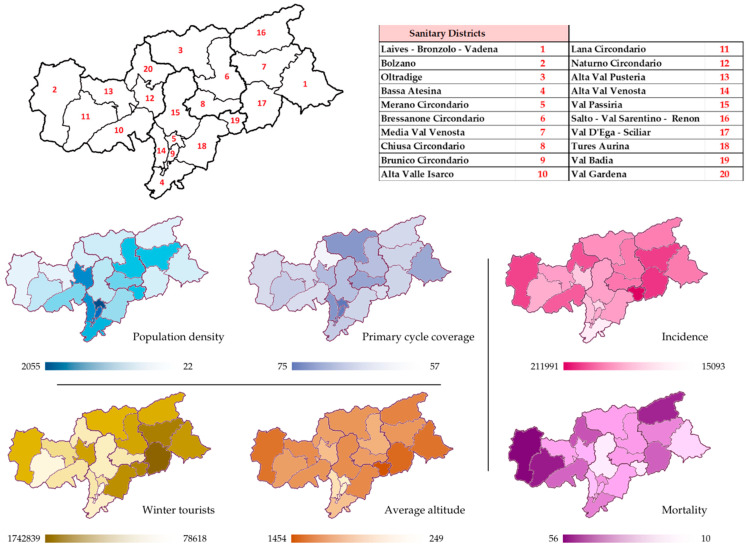
Demographic (population density—inhabitants/km^2^), healthcare (primary series vaccination coverage), social dynamics (number of winter tourists), and geographical (average altitude (m)) characteristics of the 20 districts of South Tyrol. Distributions of SARS-CoV-2 incidence and mortality are also represented and expressed as cases or deaths per 100,000 inhabitants.

**Figure 3 ijerph-21-01604-f003:**
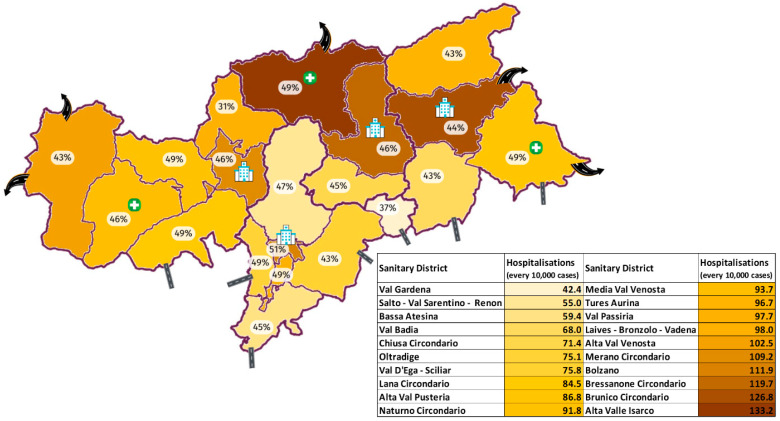
Graphical representation of the distribution of hospital admissions per 10,000 cases (brown scale) among districts of the province. Curved roads indicate connections with foreign countries, straight roads with other Italian regions. Hospitals indicate health facilities with ICUs, green crosses indicate small hospitals without ICUs, and percentages represent the districts’ vaccination coverage with booster dose.

**Table 1 ijerph-21-01604-t001:** SARS-CoV-2 incidence, hospitalization, and mortality correlation (Spearman’s Rho, *p*-value) with quantitative district characteristics of the 20 districts in South Tyrol, Italy, during winter 2021/2022.

District Characteristics		Incidence *	Hospitalization Rate *	ICU ^a^ Admission Rate *	Mortality *	Hospitalization on Cases **	ICU ^a^ Admission on Cases **	Case Fatality Rate **
Demographics	Inhabitants	−0.54 (0.015)	0.19 (0.416)	−0.27 (0.253)	−0.17 (0.478)	0.4 (0.077)	−0.23 (0.332)	−0.09 (0.696)
	Population Density	−0.44 (0.050)	−0.18 (0.435)	−0.54 (0.014)	−0.4 (0.077)	0.03 (0.905)	−0.52 (0.020)	−0.36 (0.120)
Social Dynamics	Average Salary	0.269 (0.250)	−0.114 (0.630)	0.093 (0.697)	−0.289 (0.216)	−0.205 (0.385)	0.091 (0.704)	−0.462 (0.040)
	Winter Tourism	0.54 (0.015)	0.22 (0.359)	0.29 (0.208)	−0.11 (0.645)	0.12 (0.627)	0.29 (0.218)	−0.2 (0.405)
Health Services	Primary Series	−0.62 (0.004)	0.01 (0.970)	−0.4 (0.081)	−0.43 (0.061)	0.25 (0.298)	−0.4 (0.080)	−0.36 (0.123)
	Booster Dose	−0.68 (0.001)	0.11 (0.65)	−0.38 (0.095)	−0.34 (0.139)	0.39 (0.091)	−0.37 (0.11)	−0.23 (0.326)
	Pharmacies (/km^2^)	−0.326 (0.161)	−0.095 (0.690)	−0.436 (0.055)	0.109 (0.647)	−0.424 (0.062)	−0.401 (0.080)	−0.363 (0.116)
Geography	Average Altitude	0.67 (0.001)	−0.11 (0.631)	0.23 (0.339)	0.04 (0.865)	−0.34 (0.141)	0.19 (0.420)	−0.05 (0.830)

To facilitate interpretation, cells with a *p*-value < 0.10 are colored in light grey, dark grey if *p*-value < 0.05. * Every 100,000 inhabitants. ** Every 10,000 cases. ^a^ Intensive Care Unit.

**Table 2 ijerph-21-01604-t002:** SARS-CoV-2 incidence, hospitalization, and mortality (median, IQR, *p*-value) according to qualitative characteristics of the 20 districts in South Tyrol, Italy, during winter 2021/2022.

District Characteristics		Incidence *	Hospitalization Rate *	ICU ^a^ Admission Rate *	Mortality *	Hospitalization on Cases **	ICU ^a^ Admission on Cases **	Case Fatality Rate **
Health Services	Big hospitals	17,298 (16,035–18,910)	199 (177–230)	20 (14–29)	36 (34–37)	116 (111–121)	12 (8–16)	19 (19–20)
	Small hospitals	17,825 (16,969–18,479)	160 (159–237)	5.3 (4.8–24.7)	33.9 (18.5–53)	94 (87–133)	3.1 (2.7–13.4)	19 (10–31.2)
	No hospitals	18,354 (17,365–19,661)	139 (123–168)	12.5 (0–23.1)	34.6 (31.7–43.5)	76 (68–97)	6.2 (0–12.4)	19.3 (18.5–23)
	*p*-value	0.707	0.054	0.739	0.993	0.01	0.683	0.987
Geography	Main Cities	17,298 (15,958–19,336)	199 (176–239)	19.9 (10.6–35.3)	35.5 (32.9–37.1)	116 (111–123)	11.7 (5.9–18.9)	19.1 (18.8–21.1)
	Rural/Towns	18,090 (17,240–19,510)	159 (128–173)	8.9 (1.9–23.9)	34.3 (28.9–44.4)	86 (70–97)	4.7 (1.3–12.9)	19.2 (17.2–24.4)
	*p*-value	0.617	0.029	0.554	0.963	0.005	0.437	0.963
	Bordering abroad	19,263 (18,152–20,118)	221 (183–247)	18.6 (8.7–36.4)	36 (26.2–47)	115 (95–130)	9.8 (4.5–18.6)	18.9 (14.4–23.5)
	Not bordering	17,419 (16,708–18,974)	159 (128–176)	11.3 (1.9–22.8)	34.7 (31.4–43.2)	88 (70–98)	5.8 (1.2–13.2)	19.3 (18.6–23)
	*p*-value	0.178	0.022	0.335	0.82	0.05	0.437	0.82
	Bordering Italian regions	18,479 (16,446–20,475)	132 (93–160)	4.7 (0–19.5)	33 (18.5–42.8)	75 (59–84)	2.7 (0–10.1)	18.9 (10–22.1)
	Not bordering	17,825 (17,115–18,589)	177 (160–206)	17.4 (4.8–33.7)	34.8 (33.7–45.2)	98 (94–112)	9.4 (2.7–19.3)	19.3 (18.8–23)
	*p*-value	0.643	0.006	0.275	0.351	0.003	0.211	0.351

To facilitate interpretation, cells with a *p*-value < 0.10 are colored in light grey, dark grey if *p*-value < 0.05. * Every 100,000 inhabitants. ** Every 100,00 cases. ^a^ Intensive Care Unit.

**Table 3 ijerph-21-01604-t003:** Results of the linear regressions.

Incidence	Estimate	95% Confidence Interval	Standardized Estimate	*p*-Value	Model Adjusted R^2^
Intercept	23.24	(15.13–31.34)		<0.001	0.60
Vaccination coverage (booster dose)	−0.17	(−0.32–0.02)	−0.44	0.033	*p*-value
Average altitude (km)	1.79	(−1.13–4.72)	0.34	0.212	0.001
Winter tourists (100,000)	0.12	(−0.03–0.26)	0.33	0.100	
Population density (/hectare)	0.03	(−0.13–0.19)	0.08	0.685	
**Hospitalization Proportion of Cases**	**Estimate**	**Confidence Interval**	**Standardized Estimate**	***p*-Value**	**Model Adjusted R^2^**
Intercept	73.93	(−8.63–156.48)		0.075	0.61
Vaccination coverage (booster dose)	0.29	(−1.64–2.22)	0.06	0.749	*p*-value
Hospital absence/small/big	12.37	(0.85–23.89)	0.42	0.037	0.002
Bordering abroad	17.88	(−2.16–37.92)	0.74	0.076	
Bordering other Italian regions	−20.72	(−37.05–−4.39)	−0.86	0.017	
Pharmacies density (/km^2^)	−1.67	(−88.42–85.09)	−0.01	0.968	

To facilitate interpretation, cells with a *p*-value < 0.1 are colored in light grey, dark grey if *p*-value < 0.05.

## Data Availability

The data presented in this study are available on request from the corresponding author due to privacy reasons.

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
