# Peer review of "An Ecological Study Relating the SARS-CoV-2 Epidemiology with Health-Related, Socio-Demographic, and Geographical Characteristics in South Tyrol (Italy)"

_ijerph, 2024, doi:10.3390/ijerph21121604_

Round 1

Reviewer 1 Report

Comments and Suggestions for Authors

Brief Summary: The aim of this paper was to correlate the incidence of Covid-19 infections in a region of northern Italy to several variables, including demographics, social dynamics, geography, and available health services. The main strength of the paper is that the authors examined interesting variables such as altitude and proximity to borders with other countries along with other variables with more predictable outcomes such as population and size of healthcare system. The significant contribution to the body of literature is that this particular region appears to be understudied, and conclusions from this study could help direct Covid-19 related healthcare decisions to better serve the region’s population.

General Concept Comments: The relevance of the study topic is definitely there. Some of the unique factors examined make the article interesting. The authors describe the region very well such that someone from elsewhere can envision the study area. However, I think more should be added in the introduction citing related and relevant studies. How has winter tourism affected Covid-19 infections on other regions or countries? The authors mention “inconclusive and contrasting” studies in the abstract, but I’d like more detail on some of those in the introduction. This will further help illustrate the importance of this particular study. There also needs to be more information provided about winter tourism. Are these tourists staying in the region for a weekend? Longer? Are they coming from other regions of Italy or from bordering countries? 

References are majority from 2022 and earlier. Addition of newer literature could help show that this study is still relevant even though data is from 21/22. Compare your results to more recent studies. Also lacking comparison of results to that of similar studies in terms of altitude, proximity to border countries, vaccination rates… 

Specific Comments:

-Lines 40-42 mention epidemiological characteristics differing from neighboring territories- elaborate on what these differences are/were.

-Line 94- describe “winter tourism.” What constitutes a tourist? Are they from other regions of Italy or from other countries? How is tourism measured? Does it matter how long they stayed in the region?

-Section 2.2 and 2.3- The formatting of the bullet paragraphs in these two sections do not match those in section 2.1. They should all match, and the formatting in section 2.1 is more visually appealing.

-Line 134- What is the significance of 21 days? Is there a reference for this?

-Figure 1- I think this is a useful graphic, but it is impossible to read any of the text. Way too small. Consider breaking into three separate figures and increasing the size of each.

-Lines 217-219, Table 2 caption, maybe a typo? It repeats itself, same words over twice. Confusing.

-Figure 2- I like this figure! But, again, the text in the associated table is way too small. There is plenty of space to make the text larger here.

-Lines 258-261- very long run-on sentence. Consider breaking into multiple sentences.

-Line 258- Here you used “COVID-19” where every other instance has been “SARS-CoV-2.” Keep it consistent throughout.

-Lines 259-261- regarding higher hospitalization where bordering other countries- are residents of other countries coming to Italian hospitals near the border? Need more explanation of why proximity to borders is correlated with higher hospitalization. It’s an interesting finding! And again, have other studies found this to be true?

-Line 315- Might want to consider rewording the part where you say “all” SARS-CoV-2 cases were recorded during the study period. There’s just no way that all positive cases were recorded.

Reviewer 2 Report

Comments and Suggestions for Authors

1)     Given the importance of geography to the conclusions of the study, a larger map showing the location of South Tyrol within Italy (and surrounding countries), the location of the capital city, and important physical geography would be useful to readers prior to the presentation of demographic data etc. in Figure 1

2)        Lines 52-55. It would be helpful to include the total population of South Tyrol and the Capital city Bolzano here.  While they are in Table S2, that table is not referenced until later in the paper and it would be useful to understand the scale

3)        The in-map legends in figure 1 are illegible.  Figure 1 needs significant reformatting to clearly present the relevant data.

4)        Line 209-210 contains a grammar error. “Health care facilities distribution, on the contrary, did not result associated with any significative variation of this parameter”.

5)        Line 213 should read “South Tyrol”

6)        Footnotes to Table 1 (Line 216) contain a typographic error (every 10000 Cases aIntensive Care Unit)

7)        It is unclear to me how vaccines were administered in the study region.  For instance, are pharmacies a site of vaccine administration in South Tyrol?  Some additional discussion of this issue might clarify the relevance of some of the study variables and more strongly support the conclusions.

8)        Line 243 – Figure 2 legend is still too small and difficult to read as a result.

9)        Line 287/289 – Provide a reference

10)   289-293 – effective implementation of public health measures is one possibility.  Is it also possible many cases in the previous season resulted in greater resistance/immunity within the tourist population? Are there other potential explanations? Are the tourists in this region predominantly repeat visitors each year?

11)   Line 302-306 – please clarify.

12)   Line 333 double word “in in”

13)   Line 335/336 Please expound on this statement, particularly the implications of ecological bias that can not be ruled out

14)   How strong is the relationship between winter tourism and altitude?  Does your data reflect this relationship at the spatial scales (health districts) you investigated?

15)   Supplementary Table 2 would benefit from listing the Area of each district so that the reader doesn’t  have to calculate it themselves.
